# The impacts of active and self-supervised learning on efficient annotation of single-cell expression data

Michael J. Geuenich [1,2] ✉, Dae-won Gong[1] & Kieran R. Campbell [1,2,3,4,5,6] ✉

A crucial step in the analysis of single-cell data is annotating cells to cell types and states. While a myriad of approaches has been proposed, manual labeling of cells to create training datasets remains tedious and time-consuming. In the field of machine learning, active and self-supervised learning methods have been proposed to improve the performance of a classifier while reducing both annotation time and label budget. However, the benefits of such strategies for single-cell annotation have yet to be evaluated in realistic settings. Here, we perform a comprehensive benchmarking of active and self-supervised labeling strategies across a range of single-cell technologies and cell type annotation algorithms. We quantify the benefits of active learning and self-supervised strategies in the presence of cell type imbalance and variable similarity. We introduce adaptive reweighting, a heuristic procedure tailored to single-cell data—including a marker-aware version—that shows competitive performance with existing approaches. In addition, we demonstrate that having prior knowledge of cell type markers improves annotation accuracy. Finally, we summarize our findings into a set of recommendations for those implementing cell type annotation procedures or platforms. An R package implementing the heuristic approaches introduced in this work may be found at https://github.com/camlab-bioml/leader.

Single-cell expression profiling technologies have revolutionized our understanding of healthy and diseased tissue. For example, methods that quantify gene expression such as single-cell RNA-sequencing (scRNASeq)[1] provide unprecedented insights into cellular hierarchies[1–4], differentiation[5], and rare cell types[5,6]. Similarly, technologies that measure single-cell protein expression—both in suspension such as single-cell cytometry by time of flight (CyTOF[7]) and in situ spatially such as imaging mass cytometry[8]—have uncovered cellular states and spatial architectures associated with disease progression and patient subtypes[9].

Central to understanding the high-dimensional single-cell expression profiles is the ability to categorize them into cell types and states, thus aligning them with well-grounded prior molecular biology knowledge. For example, cells may be understood by functional roles resulting in cell types such as epithelial or T-cells. Consequently, a common step in the analysis of single-cell data is to assign cells to such cell types[10]. At a more granular level, identifying certain cell states associated with disease may be of interest, such as highly proliferative Ki-67+ malignant epithelial cells across a range of cancer types[11].

Given the centrality of assigning cells to a priori specified cell types, a myriad of computational solutions have been proposed,

[1]Lunenfeld-Tanenbaum Research Institute, Sinai Health System, Toronto, ON M5G 1×5, Canada. [2]Department of Molecular Genetics, University of Toronto, Toronto, ON M5S 1A8, Canada. [3]Department of Statistical Sciences, University of Toronto, Toronto, ON M5S 3G3, Canada. [4]Department of Computer Science, University of Toronto, Toronto, ON M5T 3A1, Canada. [5]Ontario Institute of Cancer Research, Toronto, ON M5G 1M1, Canada. [6]Vector Institute, Toronto, ON M5G 1M1, Canada. ✉e-mail: mgeuenich@lunenfeld.ca; kierancampbell@lunenfeld.ca

totaling >160 for scRNASeq alone as of mid 2023[12]. Such methods and workflows may be approximately described by 3 categories. Firstly, cluster-and-interpret workflows, whereby cells are clustered via unsupervised algorithms[13,14], and clusters are interpreted as cellular types or states via either manual inspection of known markers or via automated methods[15,16]. However, these methods have been shown to underperform relative to cell-level annotation approaches[17,18]. Secondly, marker-based or "semi-supervised"[19] methods that invoke statistical models to automatically assign cells to a priori known cell types based on the over-expression of known markers[18,20,21]. Finally, there are supervised approaches that rely on previously labeled cells as training data and then treat annotation as a prediction problem[22–24]. Such approaches typically perform the best when labeled data is available and when data is labeled at the cellular level rather than cluster level[17,18].

While such supervised approaches may often leverage labels from existing atlases using transfer learning[25], such atlases are not always available or may be insufficient for a task at hand. An alternative approach to obtain a ground truth training dataset is to label a subset of the dataset manually, but this is a time-consuming process. In addition, given a large dataset comprising thousands to millions of cells, selecting a set of cells to annotate is non-trivial, as random selection would oversample abundant cell type and undersample rare cell types. A popular machine learning solution to select data points in the presence of expensive-to-obtain labels is *active learning*[26], where a model is used to suggest the next sample (cell) to label. Typically, samples with the highest predictive uncertainty are likely to maximally improve classification performance if a label is acquired. Active learning begins with an initial set of labeled data points. A classifier is trained, the predictive uncertainty in the remaining unlabeled cells is calculated, and the cells with the highest uncertainty are chosen for expert annotation. Once labeled, the classifier is retrained, and this loop continues until the desired number of cells have been annotated.

Despite the potential for active and self-supervised learning to improve single cell annotation efficiency, few studies have quantified the improvements possible through incorporating these approaches. One study compared active learning to random selection on scRNASeq datasets, finding small improvements[17]. However, the findings may not translate into real-world use cases of active learning for several reasons. For example, in this analysis a cell from every cell type was required to be in the initial annotated set, which is virtually never the case on real data since the cell types are not known in advance and it is unlikely to occur by chance due to cell type class imbalance, which is frequently high due to the presence of rare treatment resistant clones[27], immune cell subtypes[28] or progenitor cell types present low numbers[29]. In addition, no popular single-cell annotation methods specifically designed for this task were compared and the effects of various realistic single-cell scenarios like dataset imbalance were not investigated. To our knowledge, no other works exist that have examined these questions. Finally, *self-supervised* approaches such as pseudo-labelling have previously been used to boost the performance of classifiers in low-label environments[30–33] including models such as AlphaFold2[34]. However, the utility of simple self-training procedures that may improve classification efficiency has not been fully investigated.

Therefore, there are multiple outstanding questions in the utility of active and self-supervised learning approaches for cell type annotation, including (i) what performance boost do active learning approaches deliver when combined with a range of popular single-cell annotation algorithms such as SingleR[23] and scmap[22]? (ii) what (if any) active learning search strategy such as maximum entropy search performs best? (iii) what are the effects of cell type imbalance and similarity on active learning performance? (iv) can prior knowledge of cell type marker genes be used to improve the initial training set? (v) can sampling cells proportionate to their cluster identities be used as a viable alternative to active learning? and (vi) do self-training approaches like pseudo-labelling improve model performance?

Here we address these questions by performing a comprehensive benchmarking of active learning across 6 datasets, 3 technologies, 6 cell annotation methods, 24 active learning approaches and trained over 1600 active learning models. We show that active learning and adaptive reweighting—a cell selection method introduced in this work—both outperform random cell selection. In addition, we show that strategies that exploit prior knowledge of cell type markers can improve performance, and that self-supervised learning can improve annotations in various scenarios. Finally, we summarize our work with a set of recommendations for users.

## Results

### Random forest models and prior marker knowledge are best suited for active learning

We first sought to replicate existing active learning findings[17] under real world conditions. To accomplish this, we collated six single cell datasets comprising different modalities and existing ground truth labels: two scRNASeq datasets from breast[35] and lung[36] cancer cell lines where each cell line forms a "cell type" to predict, a pancreatic cancer single nucleus RNA Sequencing (snRNASeq) dataset in which cell types were assigned using a combination of clustering and copy number profiles[37], a CyTOF dataset comprised of mouse bone marrow cells[38] that was assigned using a gating strategy, a scRNASeq dataset from healthy donors of the liver atlas dataset[39] and the vasculature tabula sapiens scRNASeq dataset[40]. This covers cell type labels previously designated as gold (cell lines) and silver (gating) standards as ground truth[41]. Each dataset was subsampled to several thousand cells (see methods) for computational efficiency given that over 200,000 unique experiments were run for this analysis (S. Table 1). The resulting datasets were composed of cell types with varying similarity (Supplementary Fig. 1) and cell type imbalance (Fig. 1A). To benchmark the active learning approach, we split each dataset into ten train/test splits. To mimic the number of cells an end user would manually annotate, we selected a total of 100, 250 and 500 cells from the training set. These subsets were then used to train six cell type assignment methods using ground truth cell type labels. We then evaluated the trained classifiers using the held out test set using five different accuracy metrics (methods) (Fig. 1B). Note however, that the main goal of this work is not to evaluate the accuracies of the classifiers, but rather to benchmark the improvement in classification performance gained by creating an informative training dataset.

We first set out to replicate existing findings that random forest models outperform logistic regression active learning models[17]. Rather than ensuring one cell of each type was present in the training dataset, we randomly selected 20 cells as the initial training set for the active learning model without regard for cell type composition. Using these probabilities we selected the next set of 10 cells with maximum uncertainty at each iteration, labeled them and added them to the training set. We quantified uncertainty using two metrics: (i) the highest entropy based on the predicted cell type probabilities and (ii) the lowest maximum probability predicted for a cell over cell types. Both are well established as active learning techniques to quantify which samples (cells) a classifier is least certain about[42], and would therefore benefit most from receiving a label. While doublets were removed by the dataset authors, the approaches used for this task are generally imperfect and doublets or mislabeled cells may still exist in the ground truth dataset[43]. These would likely have the highest entropy and lowest maximum probability thus possibly corrupting the efficacy of our active learning approach. To protect against these cells being preferentially selected, we selected cells at three different certainty thresholds for each metric: cells with the highest entropy and cells that lie at the 95th and the 75th percentile of the entropy distribution and cells with the lowest maximum probability, 5th and 25th percentile of the probability distribution. This should be an effective way to ensure singlets are selected as the multiplet rate is generally

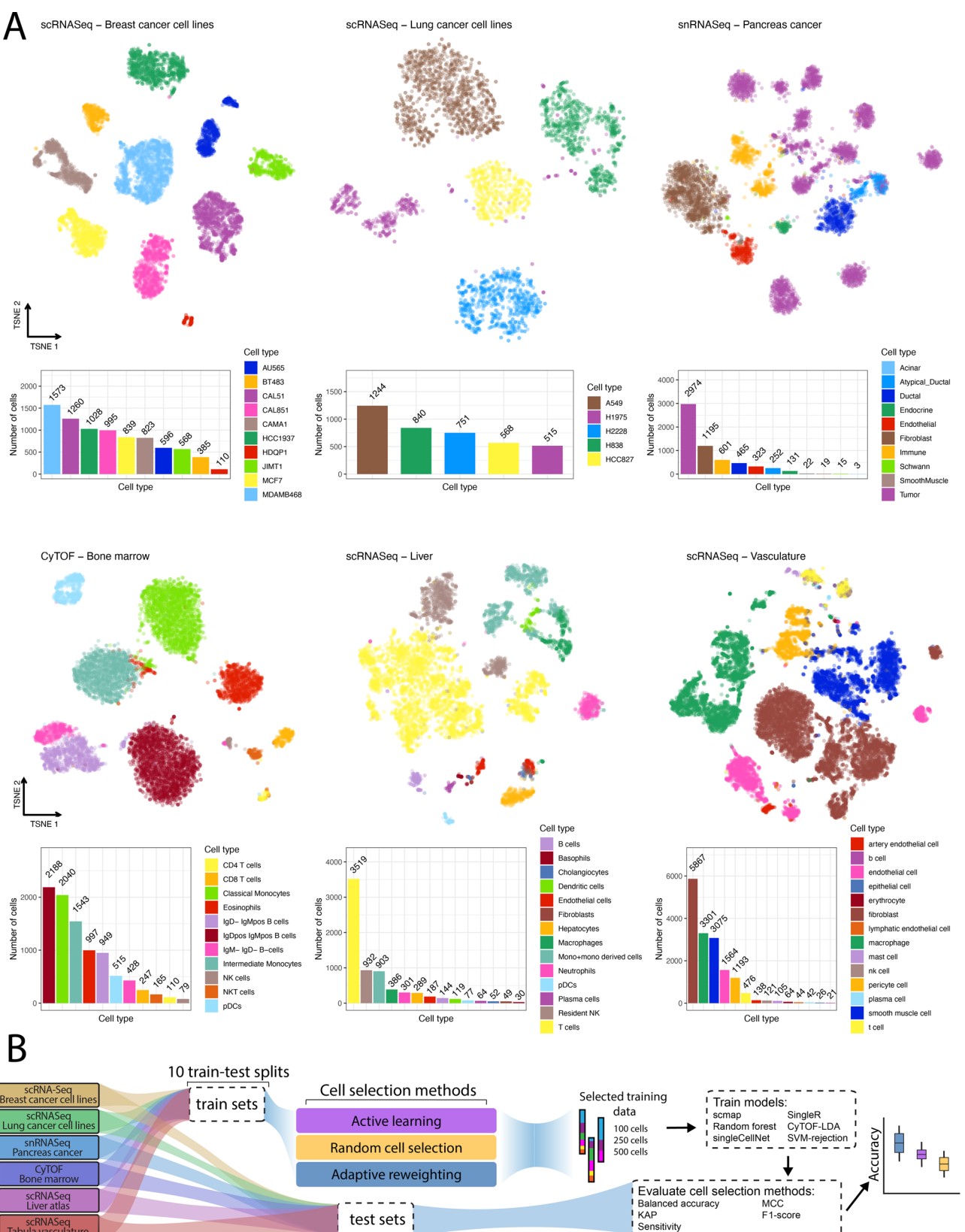

**Fig. 1 | Benchmarking overview. A** TSNE embedding of datasets used in this benchmarking colored by cell type (top) along with bar charts of cell type composition (bottom). **B** Schematic of the evaluation procedure: each dataset is split into 10 different train-test splits using a 50/50 split. Datasets of size 100, 250 and 500 cells are then sampled from the training dataset using active learning, adaptive reweighting and a random sampling (baseline). SingleR, scmap, CyTOF-LDA, a random forest model, singleCellNet, and a support vector machine is then trained using ground truth labels and evaluated by quantifying cell type prediction accuracy on the held-out test set. Source data are provided on zenodo: https://doi.org/10.5281/zenodo.10403475.

below these values[44]. Finally, to ensure our active learning methodology is valid, we calculated the performance of the active learning classifier with each iteration. This showed a steady increase in accuracy (Supplementary Figs. 2 and 3), indicating that our implementation works as intended. Using this active learning setup, we then created training datasets of sizes 100, 250 and 500 cells, and labeled these cells with ground truth labels. Using our benchmarking pipeline (Fig. 1B) we replicated existing results[17] and found that our random forest model also outperformed the logistic regression model (Fig. 2A, Supplementary Figs. 4 and 5).

Next, we explored how the initial set of cells upon which the active learning model is trained impacts performance. We hypothesized that exploiting known information about marker genes with cell type specific expression could help select the initial cells and improve active learning results. To test this, we ranked all cells by the expression of a set of cell type marker genes that were either provided by the dataset authors, derived from the data, or identified from an external database[45]. We then iterated through all expected cell types and selected the cell with the highest score for that type. We repeated this process until we selected 20 cells to serve as the initial set of cells to train an active learning model. As expected, this approach created datasets with an increased number of represented cell types relative to a random selection of cells (Fig. 2B).

When benchmarking as previously described (Fig. 1B), selecting the initial set of cells by ranking their expression resulted in an improved classification performance across datasets. This was particularly notable in situations where few cells were labeled (Fig. 2C, Supplementary Figs. 6 and 7), likely because a larger diversity of cell types is present since the initial training, which becomes less important as more cells are labeled. Overall, we replicate existing results[17] suggesting that random forest based active learning approaches outperform logistic regression in real world circumstances. In addition, we show that active learning can further be improved by selecting the initial set of training cells through a prior-knowledge informed ranking procedure.

## Marker-informed adaptive reweighting complements active learning as a cell selection procedure

Next, we considered the failure modes of existing active learning approaches on single-cell data. While active learning approaches prioritize cells with high predictive uncertainty, they require an accurate prediction model which may be difficult to achieve in certain circumstances. To address this, we developed adaptive reweighting, a straightforward heuristic procedure that attempts to generate an artificially balanced cell set for labeling. Since clusters derived from unsupervised methods are often representative of individual cell types, we hypothesized that sampling a fixed number of cells from each cluster could obtain an approximately balanced dataset with respect to the ground truth cell type labels (Fig. 3A). However, this heuristic is not perfect, as cells of a single cell type can be represented by multiple clusters. Therefore, we introduced a cell-type aware strategy that putatively assigns each cluster to an expected cell type using the average expression of marker genes (methods) and sample evenly from cell types rather than clusters. We used several clustering parameters but found no difference in their performance (Supplementary Figs. 8 and 9).

Overall, no singular method is consistently best. However active learning does outperform random selection and adaptive reweighting across most datasets, though adaptive reweighting remains competitive in some situations (Fig. 3B). Specifically, the highest entropy and lowest maximum probability selection strategies consistently outperform random cell selections. Selecting cells at the 75th and 25th entropy and maximum probability percentile threshold however consistently performed worse than random. As expected, the marker aware adaptive reweighting strategy generally outperforms the non-marker aware strategy, likely because it has access to prior knowledge in the form of marker genes. Nonetheless, care should be taken when

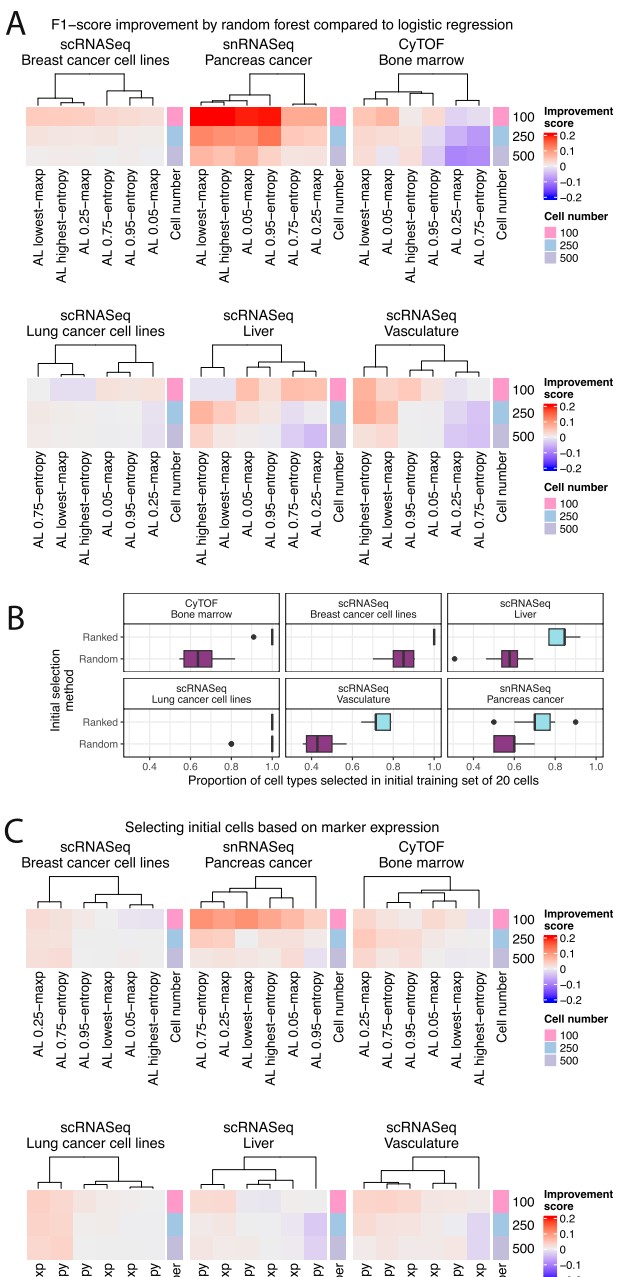

**Fig. 2 | Active learning is most effective with random forest classifiers and when selecting the initial set of cells by marker expression. A** Performance comparison of classifiers trained using a random forest and logistic regression to provide predictive uncertainty estimates for active learning. The relative difference in F1 improvement score is calculated as the difference in F1-score between the random forest and logistic regression models and standardized by the logistic regression F1-score. This score is averaged across train-test splits and cell type prediction methods. The initial set of cells was selected randomly. **B** Proportion of all cell types represented in the ground truth dataset selected by the random and ranking selection procedures for each of the 10 train test splits. Boxplots depict the median as the center line, the boxes define interquartile range (IQR), the whiskers extend up to 1.5 times the IQR and all points depict outliers from this range. **C** Same as (**A**) with the improvement score as the difference between the performance when the initial cells are selected based on cell type marker information and random selection. This score is averaged across train-test splits, active learning algorithms, and cell type prediction methods. The cell number specifies the total size of the training set. Source data are provided on zenodo: https://doi.org/10.5281/zenodo.10403475.

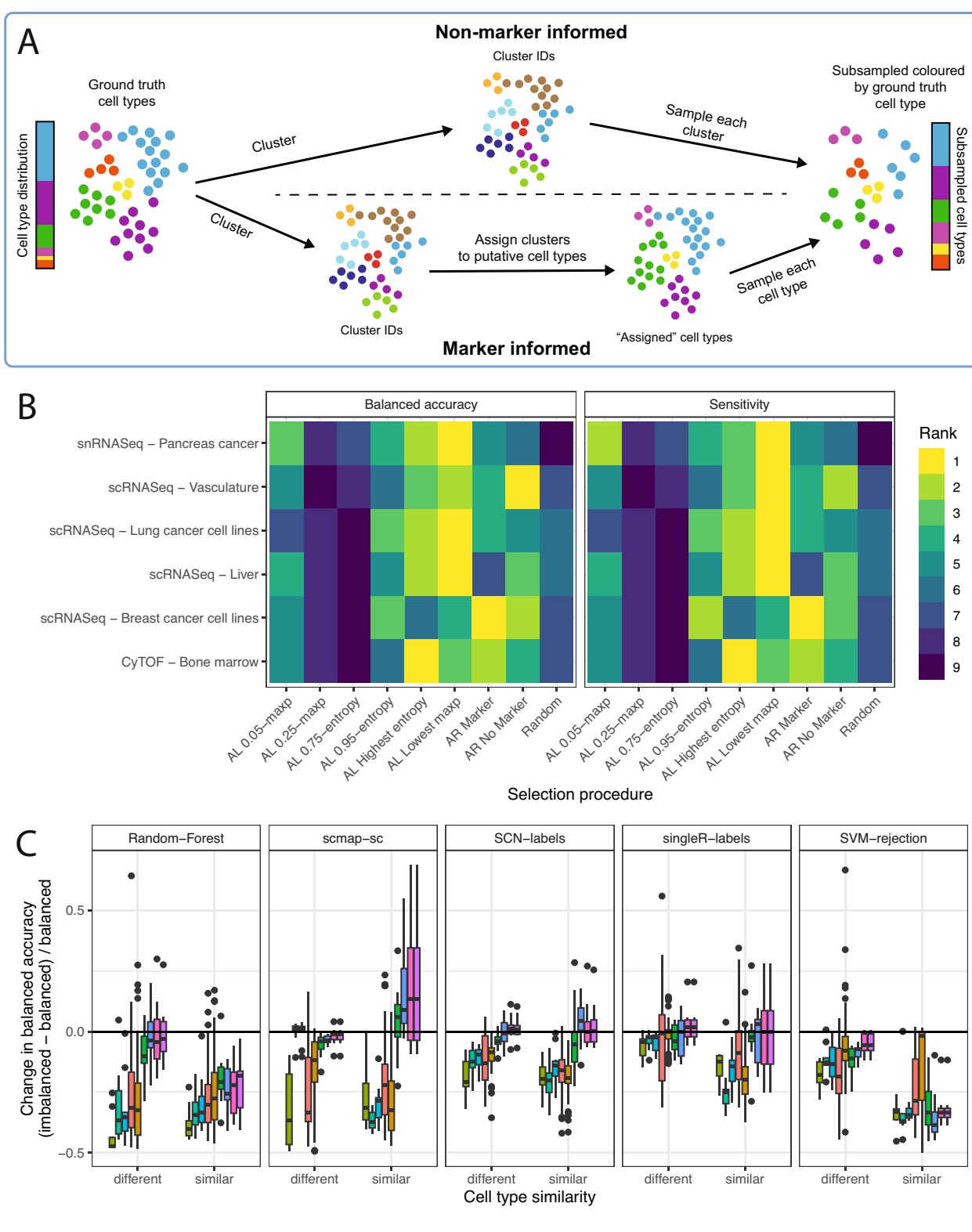

defining a set of markers, as corruption in these can lead to decreases in performance (Supplementary Fig. 10). While we focused our analysis on balanced accuracy and sensitivity for the sake of clarity, all five metrics are highly correlated (Supplementary Fig. 11). Finally, we found that no selection strategy was too run-time intensive for practical purposes (Supplementary Fig. 12).

## Active learning outperforms alternative methods in imbalanced settings

We next sought to understand the effect of cell type imbalance on active learning and adaptive reweighting given that such imbalance is common within the field of single cell biology[27], and has been shown to affect active learning results in adjacent fields[46,47]. Across

**Fig. 3 | Investigating the effect of different cell selection methods on predictive performance. A** Schematic depicting the adaptive reweighting algorithm. First, the full dataset is clustered using existing methods. In the non-marker-informed case (top) a subsampled dataset to be labeled is created by randomly selecting a set number of cells from each cluster. In the marker-informed case (bottom), each cluster is assigned a putative cell type based on the average expression of marker genes. A subsampled dataset of the size requested by the user is then created by sampling an equal number of cells from all putative cell types. **B** Performance of all selection methods tested across ten different train test splits (AL active learning, AR adaptive reweighting). Each selection method is ranked by the median balanced accuracy and sensitivity across seeds and cell type assignment methods. For the active learning results, the initial cell selection was ranked, and a random forest was used. **C** The difference in balanced accuracy between an imbalanced and a balanced dataset standardized by the balanced dataset for the snRNASeq cohort indicates improved classification accuracy by active learning approaches in imbalanced settings. Selection procedures are ordered by the average change in balanced accuracy. The balanced dataset is composed of two cell types with 250 cells each, while the imbalanced dataset is composed of 50 cells of one type, and 450 of another. In the snRNASeq cohort the similar dataset is composed of tumor and atypical ductal cells, while the different dataset is composed of tumor and immune cells. Boxplots depict the median as the center line, the boxes define interquartile range (IQR), the whiskers extend up to 1.5 times the IQR and all points depict outliers from this range. Source data are provided on zenodo: https://doi.org/10.5281/zenodo.10403475.

all datasets we sampled 450 cells of one type and 50 of another to create artificially imbalanced datasets with 500 cells. Since cell type similarity can influence performance (whereby classifying cells of similar types is harder)[48] we repeated this analysis twice: once with two distinct cell types and once with similar cell types. In the case of the snRNASeq the similar dataset was composed of tumor and atypical ductal cells for the majority and minority cell type respectively, while the distinct dataset was composed of tumor and immune cells for the majority and minority cell type respectively (Supplementary Fig. 1). In addition, we also created a balanced dataset with 250 cells of each type as a control (Table 1). Next, we sampled 100 cells from these artificially imbalanced datasets using active learning, adaptive reweighting and random selection. We then used the set of 100 cells as a training set using our benchmarking pipeline (Fig. 1B).

We found that active learning approaches generally outperformed both random and adaptive reweighting approaches in imbalanced settings (Fig. 3C, Supplementary Figs. 13–27). As expected, the drop in performance in imbalanced settings was higher for cell types that were highly similar, while it was less pronounced when the cell types selected were distinct from each other. Overall, these results indicate that active learning approaches should be considered first if there is a large suspected cell type imbalance. We next sought to understand the impact of dataset imbalance in a complex dataset with more than two cell types. We created balanced datasets containing 100 cells of five different cell types and imbalanced datasets with 400 cells from one cell type and 25 cells from four cell types (Table 2). Overall, we found active learning to also outperform other selection approaches in these settings (Supplementary Figs. 28–30).

## Active learning can identify distinct novel cell types
Next, we tested the ability of active learning approaches to identify cell types that were completely unlabeled in the initial training set. We trained our active learning models with 20 initial cells and ensured this dataset contained a specific cell type 0, 1, 2 or 3 times, while the remaining cells were selected either randomly or by ranking their expression as previously described. Upon training on this set of 20 cells, we then predicted cell type probabilities across the unannotated cells, calculated their entropies, and contextualized these values using ground truth labels.

The results show that when a cell type was excluded from the training set, the entropies for that cell type were generally higher relative to training sets that included 1, 2 or 3 cells of that type (Fig. 4A). While this increase in predictive entropy varied across datasets, it was most drastic when using logistic regression, though it was still appreciable when using a random forest classifier (Fig. 4A, Supplementary Fig. 31). In addition, even the logistic regression classifier showed little change in entropy values when some cell types (e.g. schwann cells) were removed (Fig. 4C). This is likely due to the similarity between schwann and endothelial cells (Supplementary Fig. 1), as when both were removed, schwann cells had appreciably higher entropies when no cells of this type were present than when a few were added (Fig. 4C, last panel). Based on these results we conclude that logistic regression based active learning approaches are likely to identify novel cell types quickly even if these were not selected in the initial training phase if these cell types are sufficiently distinct from one another.

## Self-training can further improve classification performance and detect mis-annotated cell types
Next, we investigated the utility of self-training—a form of self-supervised learning—to boost cell type classification performance

**Table 1 | Cell types used to generate imbalanced datasets for the first analysis considering cell type similarity**

|  | Similar | | Different | |
|---|---|---|---|---|
|  | **Majority** | **Minority** | **Majority** | **Minority** |
| scRNASeq Breast cancer cell lines | HCC1937 | CAL851 | HCC1937 | MDAMB468 |
| snRNASeq Pancreas cancer | Tumor | Atypical ductal | Tumor | Immune |
| CyTOF Bone marrow | Intermediate monocytes | Eosinophils | Intermediate monocytes | IgD+ IgM+ B cells |
| scRNASeq Lung cancer cell lines | HCC827 | H1975 | HCC827 | A549 |
| scRNASeq Liver | T cells | Resident NK | T cells | Mono+mono derived cells |
| scRNASeq Vasculature | Smooth muscle cell | Pericyte cell | Smooth muscle cell | Endothelial cell |

**Table 2 | Cell types used to generate the second set of imbalanced datasets**

|  | **Majority** | **Minority** |
|---|---|---|
| scRNASeq Breast cancer cell lines | MDAMB468 | CAL51, HCC1937, CAL851, MCF7 |
| snRNASeq Pancreas cancer | Tumor | Fibroblast, Immune, Ductal, Endothelial |
| CyTOF Bone marrow | IgDpos IgMpos B cells | Classical Monocytes, Intermediate Monocytes, Eosinophils, IgD- IgMpos B cells |
| scRNASeq Lung cancer cell lines | A549 | H838, H2228, HCC827, H1975 |
| scRNASeq Liver | T cells | Resident NK, Mono+mono derived cells, Macrophages, Neutrophils |
| scRNASeq Vasculature | fibroblast | macrophage, smooth muscle cell, endothelial cell, pericyte cell |

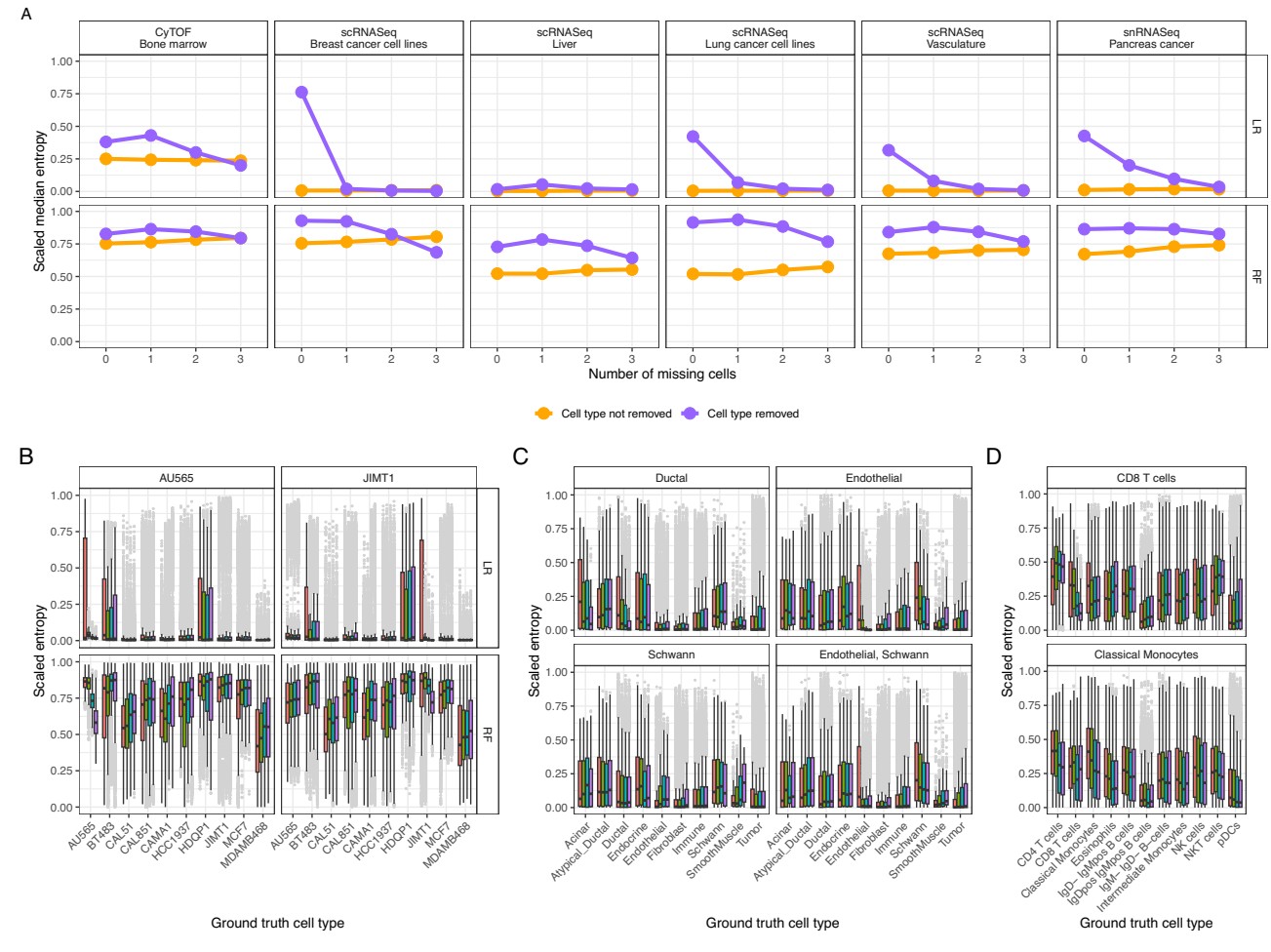

**Fig. 4 | Unlabeled cells have higher entropy values. A** Median scaled entropy values for each cohort when all cells of a type were removed (purple) and all other cells (yellow). **B** Entropy of the cell type predictive distribution for all cells not in the initial training set of 20 cells for the scRNASeq breast cancer dataset. Boxplots are colored by the number of cells present of a particular type (shown in the plot title), while the x axis shows the ground truth cell type label. **C** For the snRNASeq pancreas cancer dataset, with the bottom right panel depicting the effect of removed endothelial and schwann cells and (**D**) effect of removing CD8 T cells and classical monocytes from the CyTOF bone marrow dataset. As entropy is bounded by the total number of classes, the entropy values depicted were scaled by the maximum possible value for each experiment. Shown are the results across the 10 different train test splits. All boxplots depict the median as the center line, the boxes define interquartile range (IQR), the whiskers extend up to 1.5 times the IQR and all points depict outliers from this range. Abbreviations: logistic regression (LR), random forest (RF). Source data are provided on zenodo: https://doi.org/10.5281/zenodo. 10403475.

without requiring additional manual labeling. Self-training or *pseudo-labelling* is a technique that uses a small, labeled dataset to train a classifier that is then used to predict the label of all remaining (unlabeled) samples[49]. The most confidently labeled cells (based on the lowest entropy) are combined with the manually labeled cells to create a larger labeled dataset, which can then be used to train subsequent cell type annotation algorithms. In adjacent fields, self-training has been demonstrated to improve classification performance[34], though its efficacy for efficient cell type annotation remains unexplored.

To investigate this, we implemented random forest and logistic regression classifiers as self-training algorithms, and labeled the top 10%, 50% and 100% most confident cells with the predicted label on the three datasets from before. As expected, the accuracy of these classifiers was inversely correlated with the prediction confidence of the cells included (Fig. 5A, Supplementary Figs. 32 and 33).

We benchmarked the impact of self-training on the cell type annotation performance by combining the manually annotated dataset with a varying percentage of the most confidently labeled cells. We then used these datasets to train all the cell type annotation methods

previously implemented (Fig. 1B) and evaluated their performance on the test set. When comparing the performance of each classifier trained on only the manually annotated data to the same classifier trained on the manually annotated data including the self-trained labels, we found that in general there is an increase in predictive performance when using self-training, especially for datasets with dissimilar cell types (Fig. 5B, Supplementary Figs. 1, 34–39). The change in performance gained from self-training is most noticeable in situations with few annotated cells and is lost once 500 cells have been annotated (Supplementary Fig. 40). To further understand if any selection procedures particularly benefit from including self-training data, we correlated the classification improvement gained by adding self-trained data with the F1-score of the baseline performance achieved on only the initially labeled cells. We find that selection procedures with lower accuracies benefit most from self-training (Fig. 5C). This is likely because little performance gains can be made when a classifier already achieves a classification accuracy (F1-score) that is near-optimal.

Finally, we investigated whether self-training can be used to identify mis-labeled cells. To address this, we took a random sample of 250 cells from each of the train splits and corrupted the ground truth

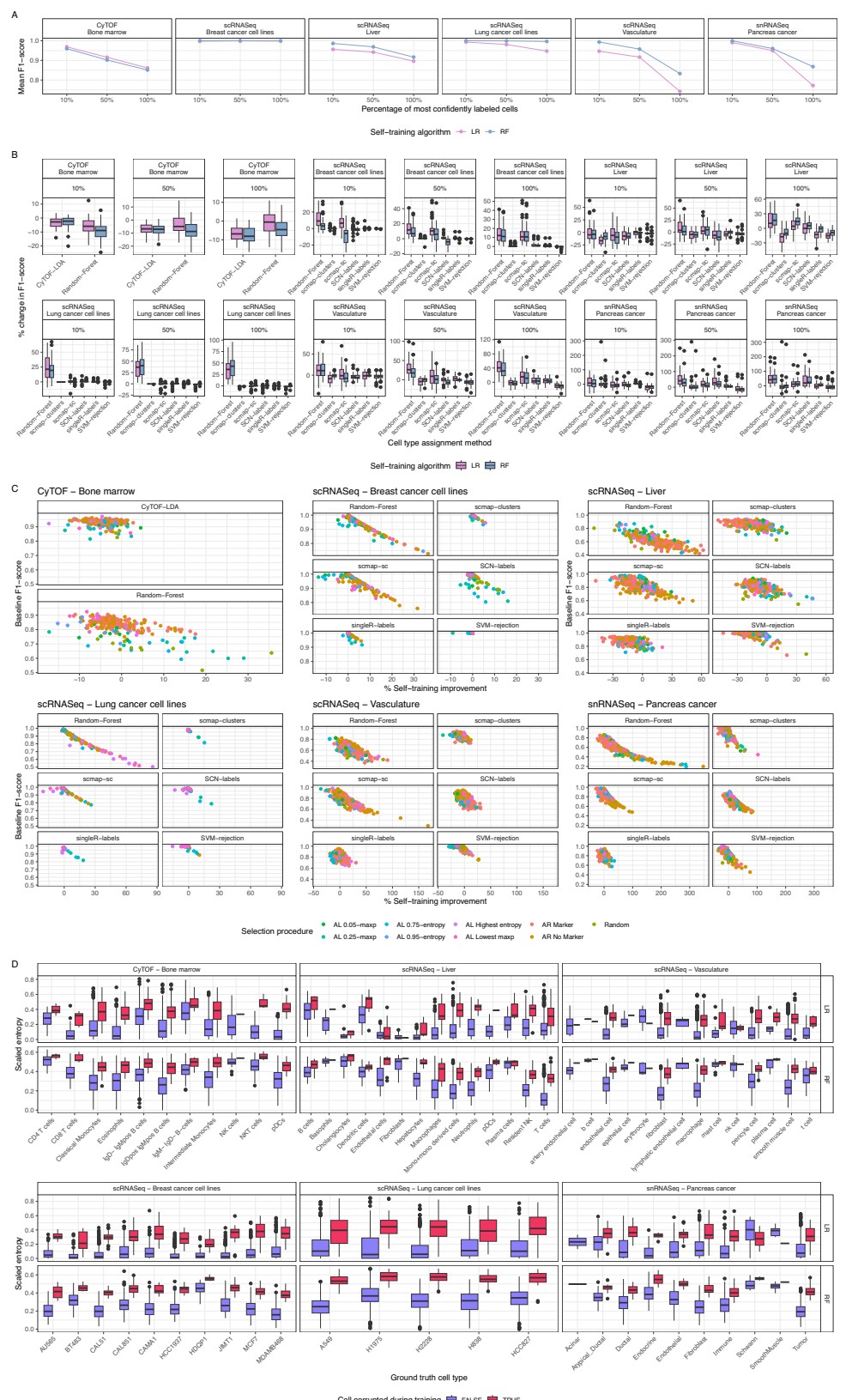

cell type for 10% of cells, such that their label was mis-assigned. We then trained logistic regression and random forest models on these 250 cells, including the misannotated ones. Next, we used this classifier to calculate the entropy for each cell in the training dataset. The results of this analysis clearly show increased entropy levels for those cells whose cell type labels were misassigned (Fig. 5D). Thus, we conclude

that self-training can also be used to detect mislabeled cells within the training set used for the self-training classifier.

## Discussion
To our knowledge, this is the first benchmarking analysis of active learning methods for cell type annotation based on real world

**Fig. 5 | Self-training can increase performance of some cell type assignment algorithms. A** F1-score of each self-training algorithm for each cohort decreases as a larger set of predictively-labeled cells are included. The x-axis shows the percent of cells with highest confidence of the overall dataset that were labeled using the self-training method. The initial set of training cells were picked using active learning. **B** Overall improvement in F1-score when including cells labeled using self-training relative to the baseline accuracy (only training on the cells labeled with ground truth values). Shown are the results using an initial training set of 100 cells that were selected using active learning and a self-training strategy. The percentage specifies the percent of most confident cells that are self-labeled. **C** Correlation between self-training improvement and original performance. **D** Entropy of all cells represented in the randomly selected 250 cell training datasets shown for each of the 10 train splits. The x-axis denotes the ground truth cell type, while each boxplot is colored by whether the cell was corrupted to a different cell type. Like Fig. 4, because entropy is bounded by the total number of classes, the entropy values depicted were scaled by the maximum possible value for each experiment. All boxplots depict the median as the center line, the boxes define interquartile range (IQR), the whiskers extend up to 1.5 times the IQR and all points depict outliers from this range. Source data are provided on zenodo: https://doi.org/10.5281/zenodo.10403475.

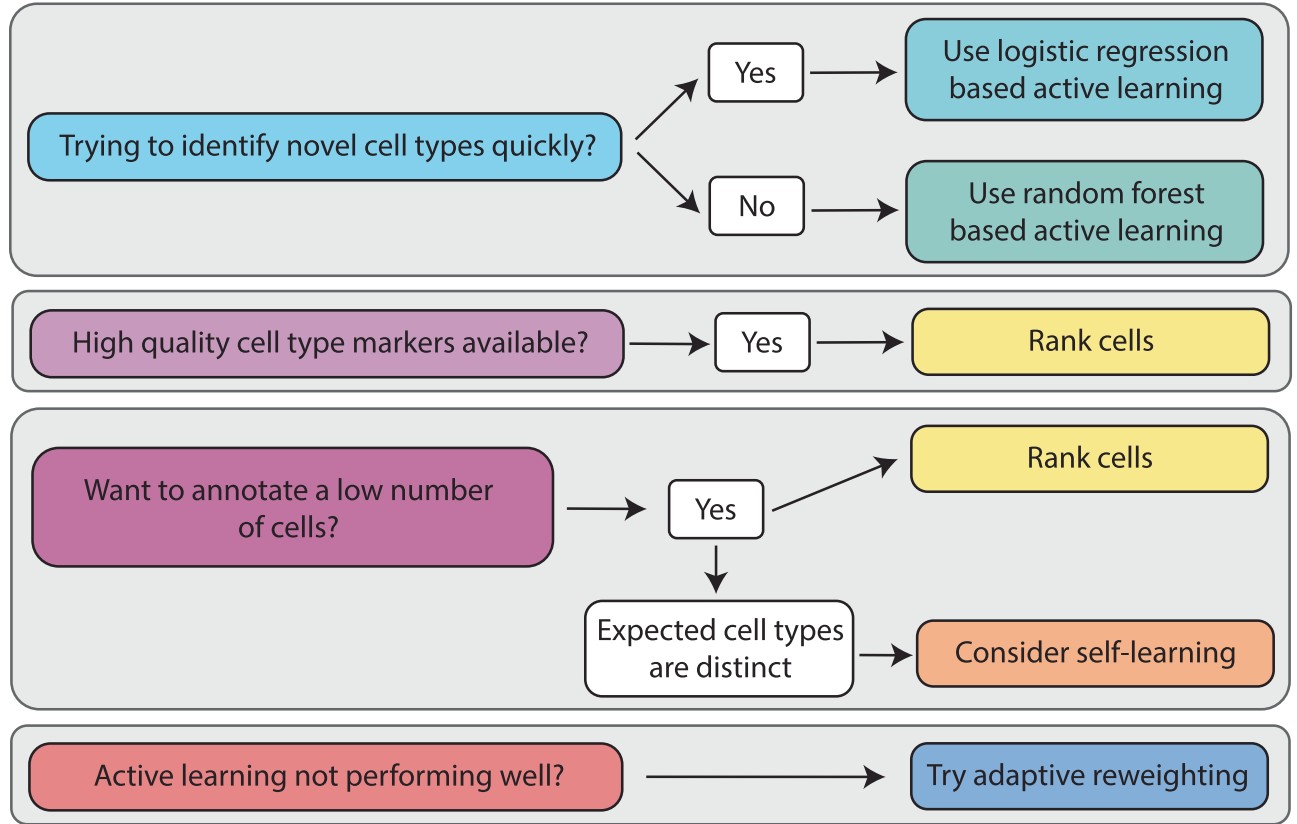

**Fig. 6 | Recommendations for practitioners.** A set of instructions outlining the criteria based on which practitioners should decide which cell type selection method to choose to aid in machine-learning based efficient annotation. The choice will heavily depend on the amount of prior knowledge available to the user in the form of possible cell type markers, dataset imbalance and the number of cells a user wishes to annotate.

assumptions. We have conducted a thorough investigation using three different modalities across six datasets, six cell type assignment methods, and have also shown that self-training can be used to boost performance. We distill the results of this benchmarking into a set of easy-to-follow guidelines for users (Fig. 6).

Nonetheless, our work has several important limitations. First, there is a tradeoff between the number of datasets used to benchmark and the comprehensiveness of the benchmarking, as doing both is very computationally expensive. As part of this work, we chose to benchmark many hyperparameter and cell type annotation tools at the expense of investigating a large number of datasets. Secondly, we relied here on "ground truth" cell type labels derived from previous studies. While we attempted to address this by using a mixture of "gold" and "silver"-standard labels as introduced in previous studies[41], there is still a fundamental trade-off of defining exactly what the ground-truth of a cell type label means in single-cell biology. While in practice a user would manually annotate cells, relying on manual annotation for this work was not feasible due to the large number of experiments performed. Instead, we simulated the manual annotation process using existing cell type annotations. There is also circularity in the way some of our cell type markers are defined, especially for the cell line data, as these are derived from a differential expression between the cell lines. However, there is a tradeoff for this part of the analysis: while well validated marker genes exist for "real" datasets, no ground truth cell type labels exist, and conversely for cell line data, ground truth "cell types" exist but no cell type markers. To minimize the bias this induces, for other experiments we used markers from a third-party database[45] where available and benchmarked the effect of marker corruption on adaptive reweighting (Supplementary Fig. 10). Finally, we note that our benchmarking approach to cell type assignment relies on labeling individual cells rather than clusters of cells. We chose this approach as it has previously been shown to improve classification accuracy[17,18]. This is likely due to imperfect clustering leading to erroneous cell type assignments and difficulties in setting cluster resolution to match expected cell types exactly. However, such a cluster-based approach would ameliorate assignment issues due to

dropout where individual genes are detected in one cell, but not another[50] due to low sequencing depth. This phenomenon could complicate cell type annotation at the single cell level if specific cell type markers are not expressed.

Future work to extend from this study could focus on several directions. For example, there is scope to extend this benchmarking across many additional datasets and modalities. In addition, an accessible web platform to enable manual annotation of cells is required for this work to be easily translatable to end users. To address the dropout issue while maintaining the accuracy benefit of labeling individual cells, hybrid approaches could be developed that employ active learning and clustering. For instance, meta-cells consisting of the combined expression of a small number of highly similar cells could be labeled, thereby reducing the chance of erroneous clustering while blunting the dropout issue. We also envision space for future work on increasing labeled datasets with semi-supervised learning approaches. Specifically, future work could focus on identifying the best certainty threshold to select cells to label using self-training. Finally, while we have shown that mislabeled cells have higher entropies, we only used standard machine learning tools and expect that approaches tailored to single cell data may outperform our implementations.

## Methods

### Adaptive reweighting
The goal of adaptive reweighting is to generate a balanced dataset by selecting a specified number of cells from a larger dataset without requiring any cell type labels. Thus, rather than sampling cells randomly, adaptive reweighting attempts to sample cells from each cluster to get an even number of cells from all cell types. To accomplish this, we cluster with Seurat[13]. Next, we sampled cells from the resulting clusters such that the same number of cells is taken from each cluster. Specifically, the total number of cells requested is divided by the number of clusters to get the total number of cells to sample from each cluster. Should a cluster have fewer cells than this, all cells from this cluster are sampled and the remaining cells are taken evenly from other clusters.

To further boost performance we also implemented a cell type marker aware version of adaptive reweighting. This variation performs the same type of clustering already described, but instead of sampling from the clusters directly it first attempts to assign the most likely cell type to each cluster. Specifically, the average expression of a set of marker genes provided by the user is calculated for each cell type and cluster individually. This process is repeated for all the cell types listed by the user resulting in a cluster by expected cell type matrix containing the average expression of marker genes for that cell type in that cluster. If negative selection markers are provided, their average expression is also calculated and subtracted from the average positive marker expression. Finally, each cluster is 'assigned' the expected cell type for which it has the highest enrichment score. To get a set of cells to annotate, cells are randomly sampled in equal numbers from the 'assigned' cell types. If all cells from an 'assigned' cluster are sampled but more cells are requested, the remaining cells are sampled at equal proportions from the other 'assigned' clusters. As part of the benchmarking for this work, Seurat was run using the first 30 principal components, resolution parameters of 0.4, 0.8 and 1.2, and nearest neighbor parameters of 10, 20 and 30 were tested. However, no meaningful differences were found between the different parameters.

### Datasets and data processing
Six datasets with publicly available ground truth cell type labels were used in this study: four single cell RNA-Seq, a single nucleus RNA-Seq, and a CyTOF dataset. The first single cell RNA-Seq dataset was composed of breast cancer cell lines[35] downloaded from https://figshare.com/articles/dataset/Single_Cell_Breast_Cancer_cell-line_Atlas/

15022698 and was randomly subsampled to 10 cell lines for computational efficiency, the second RNA-Seq dataset was composed of five lung cancer cell lines[36] downloaded from https://github.com/LuyiTian/sc_mixology. We selected cell lines as they have previously been used as a surrogate for cell type in benchmarking analysis as a "gold standard" ground truth[41]. The tabula dataset[40] was downloaded from https://figshare.com/articles/dataset/Tabula_Sapiens_release_1_0/14267219, the liver dataset[39] was downloaded from https://www.livercellatlas.org and the nuclear RNA-Seq dataset composed of untreated pancreatic cancer patient specimens comprising 11 cell types[37] was subsampled to 6,000 cells and downloaded from https://singlecell.broadinstitute.org/single_cell/study/SCP1089/human-treatment-naive-pdac-snuc-seq. The CyTOF dataset is composed of 24 cell types from the bone marrow of 10 different mice[38]. It was downloaded using the HDCytoData R package[51] and the most abundant cell types from a 10,000 cell subsample were used as the final dataset. Any cells annotated as being of an unknown type were removed. Each dataset was then randomly split into ten different train and test datasets using a 50:50 split while stratifying by cell type to retain the original dataset imbalance. As recommended[51], the CyTOF dataset was arcsinh transformed using a cofactor of 5. For the scRNASeq and snRNASeq datasets we calculated logcounts and normalized with the size factor using the scuttle R package[52].

For the cell line datasets, marker files were generated by conducting a differential expression analysis between a cell line and all others using the findMarkers function from scran[53]. While this is circular reasoning it is the only way to derive cell line marker genes as markers do not exist the way they do for "real cell types". A marker file for the snRNA-Seq dataset was adapted based on information provided by the authors (personal communication), while the marker file for the CyTOF dataset was adapted from the gating strategy used by the dataset authors. To avoid circularity, the marker files for the tabula datasets were created based on the 20 most sensitive markers from the PanglaoDB database[45].

### Cell type similarity calculation
To quantify the similarity between cell types we calculated the weighted cosine distance between all cell types in PCA space. Specifically, we calculated a PCA embedding for each cohort and used the first 20 principal components to calculate the cosine distance weighted by the variance explained by each principal component between the center (average) representation of each cell type and all other cell types.

### Statistics and reproducibility
No statistical method was used to predetermine sample size. No data were excluded from the analyses. The experiments were not randomized. The Investigators were not blinded to allocation during experiments and outcome assessment.

### Benchmarking cell selection approaches
We benchmarked three cell selection methods: random selection, active learning approaches, and adaptive reweighting, a method introduced in this work. Using each approach we selected 100, 250 and 500 cells from the training dataset to be labeled. The selected and labeled cells were then used to train an array of cell type prediction methods that were subsequently evaluated using the test dataset. We repeated this procedure 10 times with 10 different train test splits.

### Initial cell selection procedures for active learning
To implement an active learning approach an initial set of cells needs to be labeled to train a model that can then suggest the next set of cells to label. We tested two approaches: in the first a random set of 20 cells is selected and annotated. In the second approach, cells are first scored by the average expression of the respective marker genes for the set of

pre-specified cell types. Then the highest ranked cell for each cell type is selected one at a time until 20 cells have been selected (e.g. the highest ranking cell for cell type one is selected first, then cell type two, etc. If necessary this process loops and selects multiple putative cells from the same cell type).

## Active learning strategies

Active learning is based on the idea that the cells a classifier is least certain about are most useful in increasing the predictive accuracy when labeled. Upon having the initial set of labeled cells, we trained a random forest and a logistic regression classifier using the first 20 principal components of scaled and centered logcount expression data as input using the caret R package[54] with default parameters. The trained classifier is then used to predict across all unlabeled cells, and the predicted probabilities are used to quantify the uncertainty of the predictions for each individual cell. Out of the many active learning sampling strategies proposed[55] we chose to evaluate a maximum probability and an entropy-based method. The maximum probability measure is simply the highest predicted probability of each cell, with the lowest values corresponding to the least certain predictions. The entropy $H$ for each cell is calculated as follows:

$$H(p) = -\sum_{i=1}^{C} p_i \log_2(p_i) \tag{1}$$

Where $C$ is the set of cell types defined in the marker file by the user, and $p$ is the probability that a cell is of cell type $C_i$. Here, the highest values correspond to the least certain predictions.

Since the number of cells annotated with each iteration has been shown to not have a meaningful effect on performance[17], we selected the set of 10 cells with the highest uncertainty to label using ground truth data. After this set of cells is labeled, the classifier is trained again and the loop repeats. Since there may be doublets or even some mislabeled cells in the ground truth dataset, these would likely have the highest uncertainty thus possibly corrupting the efficacy of our active learning approach. To protect against these cells being preferentially selected, we selected cells at three different certainty thresholds for each metric: cells with the highest entropy and cells that lie at the 95th and the 75th percentile of the entropy distribution (corresponding to the lowest maximum probability, 5th and 25th percentile of the probability distribution). This should be an effective way to select singlets as the multiplet rate is generally below these values[44].

## Implementation of cell type prediction methods

To benchmark each cell type selection method we trained several cell type prediction methods on the generated training sets, and tested their accuracy on the held out test set. For all transcriptomic methods we implemented singleR[23] and SingleCellNet[56] using default parameters, SVMpred as previously implemented[57] and sc-map[22] by projecting to individual cells (scmap-cell) and clusters (scmap-clusters). For the CyTOF dataset we implemented CyTOF-LDA[24], which was specifically developed for cytometry data using default parameters. In addition, we also implemented a custom random forest predictor. For this method, logcount or arcsinh transformed expression values are scaled and PCA transformed prior to training. The classifier is trained using five fold cross validation, and the parameters explored are 4, 6 and 10 maximum features, 100 and 150 estimators as well as 25 or 50 principal components for the transcriptomic datasets and 20 or 39 (the total number of markers) for the CyTOF dataset. Each model was then evaluated using a total of five metrics: sensitivity, F1-score, Matthew's correlation coefficient, Cohen's kappa and balanced accuracy. Any cells predicted as 'unassigned' were removed prior to calculating these metrics.

## Adaptive reweighting marker corruption

To measure the effect of mis-specified markers on adaptive reweighting, we took the original marker file defined for each dataset and randomly corrupted 10, 25, 50, 75 and 100% of markers. Each of these randomly selected markers was replaced with a random gene not in the original marker file that was also among the 10,000 most highly expressed genes in the dataset.

## Creating imbalanced datasets

To benchmark the effect of cell type imbalance on the performance of selection methods, we created four datasets per cohort. Specifically, we created two imbalanced datasets with 450 cells of one type and 50 of another, with cell types that were similar and distinct from each other. We also created two additional balanced datasets as a comparison using the same cell types. The exact cell types used in each case are shown in Table 1. Only two cell types were included in this analysis to ensure we could control for cell type similarity.

To generate a more realistic scenario with more than two cell types we generated another set of balanced and imbalanced datasets for each cohort. Specifically, we randomly selected 500 cells with 100 from the majority cell type and 25 from each minority cell type for the imbalanced dataset and 100 cells from each cell type for the balanced dataset (Table 2).

## Self-training

Inspired by AlphaFold2[34], we trained a logistic regression and random forest model on the labeled cells and predicted the cell types of the remaining cells in the training dataset. We then selected the top 10, 50 and 100% most confidently predicted cells based on their entropy and labeled these using the prediction values obtained. These predictively labeled cells were then combined with the originally labeled cells and used as a training set for the cell type prediction methods. The effectiveness of self-training was measured by comparing the accuracy of predictions on the test set when a cell type prediction method was trained on the originally labeled data with or without the self-trained data.

## Reporting summary

Further information on research design is available in the Nature Portfolio Reporting Summary linked to this article.

## Data availability

All relevant data supporting the key findings of this study are available within the article and its Supplementary Information files. All data used as part of this work is publicly available from the cited studies. The scRNASeq breast cancer dataset was downloaded from: https://figshare.com/articles/dataset/Single_Cell_Breast_Cancer_cell-line_Atlas/15022698. The scRNASeq lung cancer cell line dataset was downloaded from: https://github.com/LuyiTian/sc_mixology. The scRNASeq tabula sapiens vasculature dataset was downloaded from: https://figshare.com/articles/dataset/Tabula_Sapiens_release_1_0/14267219. The scRNASeq liver cell atlas dataset was downloaded from: https://www.livercellatlas.org. The pancreatic cancer snRNASeq dataset was downloaded from: https://singlecell.broadinstitute.org/single_cell/study/SCP1089/human-treatment-naive-pdac-snuc-seq, and the CyTOF bone marrow dataset was downloaded using the HDCytoData R package. Cell type markers were either taken from the supplementary data of the original publication or downloaded from pangaloDB at https://panglaodb.se. Source data to re-create figures has been deposited on zenodo: https://doi.org/10.5281/zenodo.10403475.

## Code availability

All code required to reproduce this study can be found at https://github.com/camlab-bioml/active-learning-benchmarking and https://

doi.org/10.5281/zenodo.10397828[58]. All software packages used are listed in the Docker and Pipfile on the GitHub repository and on Zenodo.

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

## Acknowledgements

We would like to thank William Hwang and Carina Shiau for providing us with a cell type marker file for their dataset. We would also like to thank the reviewers for their effort, time, and suggestions to improve our work. This work was supported by funding from CIHR project grant PJT175270 (KC), NSERC Discovery grant RGPIN-2020-04083 (KC), a CFI/JELF award (KC), a Doctoral Student Fellowship from the Toronto Data Science Institute (MG), a Health Informatics and Data Science award from the Terry Fox Research Institute (MG) and support from the Princess Margaret Cancer Foundation (MG). This research was undertaken, in part, thanks to funding from the Canada Research Chairs Program.

## Author contributions

Project conception: MG, DG, KC. Result interpretation and manuscript writing: MG, KC. Data analysis and coding: MG.

## Competing interests

The authors declares no competing interests.
