## [Peer Review File · Nature Communications]

The impacts of active and self-supervised learning on efficient annotation of single-cell expression dataReviewer #1 (Remarks to the Author):

The authors focused on cell type annotation using active learning and self-supervised strategies. The article is extensive in workload and discusses the case of balanced and unbalanced data using multiple approaches. There are also the following issues that need to be modified before the paper can be accepted.

1. heuristic procedures are mentioned in the author's abstract, but they only appear in the abstract and are not found elsewhere in the paper. Add an explanation of this method in the paper
2. The author only tests on several existing models and does not propose his own unique model, which can only be regarded as a summary proof work. Is it possible to propose models in this respect that are more appropriate for most tasks
3. The setting of parameters of different models will also affect the results of different data sets, so multi-parameter experimental comparison should be set
the author confirms in the Introduction that "The effects of various realistic single-cell scenarios like dataset imbalance were not investigated". Most of the current research in this field is unbalanced.
4. In the classification task, imbalanced data sets have other evaluation indicators in addition to F1-score, such as AUC or MCC. (Bib, 2022, bbac524; doi.org/10.3389/fgene.2022.1038919)
5. What is the principle of extracting 100,250,500 cells?
in the section of "Active learning outperforms other methods in imbalanced settings", in order to compare the performance of active learning in imbalanced dataset with other methods, an unbalanced data set is artificially constructed. However, there are only two cell types in the unbalanced data set, and the data set in reality contains more abundant data types. It is suggested to construct the unbalanced multi-type data set to better simulate the real situation

Reviewer #2 (Remarks to the Author):

Summary

The authors introduce relatively new machine learning methods (active learning and self-supervised training based on a subset of manually labeled data) into the domain of single cell RNAseq label assignment. They found that random forest outperforms logistic regression approaches both in accuracy and in robustness in to label imbalance. They showed that active learning, and focusing on low entropy cells can boost performance. They further interrogate the impact of label imbalance and utilize adaptive weighting for under-representing over-abundant classes in the training set. The authors also make use of normalized entropy measures of marker genes within clusters as a measure of label confidence, which is a nice approach. I'd also commend the authors on their depth over breadth approach in benchmarking, as well as the acknowledgement of the intrinsic limitations of benchmarking with single cell datasets, and the caveats with ground-truth in single cell biology. However, I do have major concerns that I think would improve the manuscript if addressed.

Major

1. Are the proposed methods suited for transfer learning (building a classifier based on a prior dataset), or in the proposed methods, is there still the necessity of providing manual labels to a subset of cells from the different those found either through the active-learning process or unsupervised analyses? From my reading of the benchmark description at the bottom of page 4 ("These subsets were then used to train four cell type assignment methods using ground truth cell type labels"), it sounds like the previously designated ground truth labels were used in the training process, which may not reflect the typical process in which a user will need to actively define a new label for their newly collected dataset, given that no prior ground truth will be present for a new dataset. If the intended method can be easily applied to a scenario in which the user obtains a new, completely unlabeled dataset, the manuscript would benefit from demonstrating this (although perhaps only for a single use show-case of active learning, rather than sweeping through all related methods discussed).
2. The specific methodology used for 'known marker gene' usage may not be appropriate, if I

understand the approach correctly. On page 5, the authors state "To test this, we ranked all cells by the expression of a set of cell type marker genes provided by the dataset authors." In essence, under this paradigm, we have ground truth labels, generated by the original authors, then gene-sets that were derived downstream of these cluster assignments through differential expression. The authors then use these known differentially expressed genes (as previously determined in this dataset), to identify the cluster labels that were used to perform the differential expression analysis that yielded the marker genes; this seems a bit logically circular. Can the authors clarify this point? If my original interpretation is correct, I believe that a better test of utilizing prior expected marker genes that would be more reflective of a real-world scenario would be utilizing previously published, orthogonally collected, marker genes from a marker gene database (so long as they are careful to ensure the datasets used for testing the method did not contribute to the marker gene database). The same comment may apply to other analyses if I have correctly understood the method as currently employed.

3. On page 7, the authors state that "adaptive reweighting outperforms average selection and active learning across all accuracy measures and datasets, though with high variability (Fig. 3B)." It is a bit difficult to tell from the provided plots, given this variability. Could the authors please provide something like an average rank performance across all the tested datasets for each measure to more clearly display this?

4. Similar to the above, it is a bit difficult to clearly see in Figure 4, that having 0 cells in training increases entropy. Perhaps the authors could provide a summary panel, with the effect of ground truth cell type regressed out, as this also seems to have a large effect. As an aside – I agree with the authors' intuition that the cell-type specific effect of why a given type's absence may have variable effects on classifier performance (as with the schwann/endothelial example).

Minor

1. There may be an incomplete sentence, or extra word on page 5: "..., and would therefore benefit most from receiving a label of." Perhaps the word "of" should be removed?

2. In the legend of Fig. 2, the word size appears twice in close succession: "The cell number specifies the total size of the training set size."

Reviewer #3 (Remarks to the Author):

Geuenich and colleagues performed a comprehensive benchmarking of active and self-supervised learning strategies for cell type annotation. The observation is that prior cell type markers knowledge is useful and a marker-aware adaptive reweighting strategy is helpful. The paper is clearly writing and could offer insights for the development of cell type classification tools and applying existing tools to a new dataset. However, to reach the current conclusion, I think more experiments are needed, specially the including of more datasets.

1. The current pipeline is only tested on 3 datasets and 5 cell type classification methods. In contrast, the conclusion and contribution made by this paper is the claim that some techniques are useful in general, rather than proposing a new method. Therefore, I think it is necessary to benchmark on more cell type assignment methods and datasets.

2. More cell type assignment methods:

<https://genomebiology.biomedcentral.com/articles/10.1186/s13059-019-1795-z>

3. More datasets. I think it is necessary to test on datasets with more cell types, including Tabula Muris, Tabula Drosophilae, Tabula Muris Senis, and Tabula Sapiens.

4. Evaluate the cross dataset setting. Real-world cell type assignment is often in a cross-dataset setting, where the model is trained on one dataset and then used to classify cells in another independent dataset. This would involve batch correct and mapping of cell types. The noise in marker genes and annotated labels in the training data might also affect the performance on the test dataset. It is therefore necessary to evaluate this setting.

5. Could you comment on the running time of using these additional techniques? Will that make the training time substantially longer?

6. Noise in the cell type marker file. Cell type marker genes could be noisy. It might be interesting to randomly add noise to the makers and see if the proposed method is robsut to such noise.

We thank all three reviewers for the time they have taken to provide feedback on our work and in helping us improve our manuscript.

Reviewer 1

1. heuristic procedures are mentioned in the author's abstract, but they only appear in the abstract and are not found elsewhere in the paper. Add an explanation of this method in the paper

We apologise for the confusion caused in this respect. Adaptive reweighting, the method introduced in this paper, is the heuristic procedure mentioned. We have clarified this in the abstract and the main text.

2. The author only tests on several existing models and does not propose his own unique model, which can only be regarded as a summary proof work. Is it possible to propose models in this respect that are more appropriate for most tasks

We agree with the reviewer that we do not propose new *cell type assignment* models. However, to aid practitioners in the generation of ground truth cell type labels, we propose adaptive reweighting, a new method to select a set of cells from large datasets.

In addition, we would like to note that the performance of any given classifier is not our main concern in this work. We are predominantly interested in the boost in performance that can be gained from an existing cell type selection method by selecting cells to label using active learning methods (e.g. relative to a random selection of cells). We have further clarified this point in the manuscript.

3. The setting of parameters of different models will also affect the results of different data sets, so multi-parameter experimental comparison should be set the author confirms in the Introduction that "The effects of various realistic single-cell scenarios like dataset imbalance were not investigated". Most of the current research in this field is unbalanced.

We apologise for not having made it clearer that this analysis is indeed included in the manuscript. Below we have added a summary of all the hyperparameters explored, we have also amended the manuscript to include this table in the supplementary material.

Thus, we have explored a hyperparameter space of 168 parameters for each train test split. Since there are 10 train test splits, more than 5,000 jobs have to be run for each dataset. This is then repeated for the imbalance and self-learning analysis over an additional set of parameters. Given the request for further datasets and cell type annotation methods we have been reluctant to include even more parameters in our analysis. Nonetheless, if there are any other specific parameter configurations that would be of interest, we would be happy to explore them.

In addition, here too we would like to note that our aim was not to identify the best performing hyperparameter configuration for the classifiers implemented. Rather, the main aim was to measure how much classifier performance can be improved with better training datasets over a reasonable set of possible configurations.

	Parameter	Possible values	Number of configurations tested
Models			
Scmap	Classification level	 ● Cell level ● Cluster level 	2
SingleR	Default		1
RandomForest	Internal grid search over parameters		1
CytoTOF-LDA	Default		1
SVM-rejection	As described in (Abdelaal et al., 2019)		1
SingleCellNet	Default		1
Subtotal			7
Cell selection			
Active learning	Initial cell selection	 ● Random ● Ranking 	2
	Entropy uncertainty	 ● Highest entropy ● 95% entropy ● 75% entropy 	3
	Maxp uncertainty	 ● Lowest maxp ● 5% maxp ● 25% maxp 	3
Subtotal			12*
Adaptive reweighting	KNN parameter	 ● 10 ● 20 ● 30 	3
	Clustering resolution	 ● 0.4 ● 0.8 ● 1.2 	3
Subtotal			9
Random selection	Repeated 3 times		3
Subtotal			3
Number of cells selected		100, 250, 500	3
Total			504**

* The total is 12, as for each entropy and maxp selection metric each cell selection is run, thus the total is $3 \times 2 + 3 \times 2 = 12$.

** The total number of jobs per dataset split is the number of cell selection methods times the number of models fit times the number of cells selected: $((12 \times 7) + (9 \times 7) + (3 \times 7)) \times 3 = 504$.

Finally, regarding the imbalance section highlighted by the reviewer: the specific sentence in question refers to the state of the field investigating active learning for single-cell analysis prior to our work. We took cell type imbalance into account in three different ways: first we retained the imbalance nature of all datasets, and as such this is implicitly contained within the benchmarking shown in figure 3. Second, we created artificially imbalanced datasets of two cell types and evaluated the performance of all selection methods on these datasets (see figure 3C). Finally, as part of our edits as part of this review and inspired by the final comment of the reviewer we created an additional synthetic dataset with more than two cell types.

4. In the classification task, imbalanced data sets have other evaluation indicators in addition to F1-score, such as AUC or MCC. (Bib, 2022, bbac524; doi.org/10.3389/fgene.2022.1038919)

We apologise to the reviewer for not having included these results in our manuscript. We have now added these for all cohorts, cell type assignment methods and metrics (S. Figure 13-27). Overall, our conclusions hold across all five metrics evaluated.

5. What is the principle of extracting 100, 250, 500 cells? in the section of "Active learning outperforms other methods in imbalanced settings", in order to compare the performance of active learning in imbalanced dataset with other methods, an unbalanced data set is artificially constructed. However, there are only two cell types in the unbalanced data set, and the data set in reality contains more abundant data types. It is suggested to construct the unbalanced multi-type data set to better simulate the real situation

In regards to the principle of extracting 100, 250 and 500 cells: this is the number of cells we assume the user will label. In the machine learning literature this is known as the label budget (Ren et al., 2021). From past experience we have found this number to be a good range to balance accuracy with time/effort spent labelling (Geuenich et al., 2021).

Nonetheless this is a parameter that can be adapted by the user as necessary. For context if it takes a user 10 seconds to label the average cell, labelling 100 and 500 cells would take 16 minutes and one hour and 20 minutes respectively.

We agree with the reviewer that in reality, datasets are highly imbalanced and composed of more than just two cell types. The impact of this type of imbalance on performance however is captured by our main analysis (Fig. 3) since all datasets are imbalanced (Fig 1A). In addition, cell type similarity is a confounder, as cell type classification in datasets with very different cell types is much easier than in datasets with highly similar datasets (this is also the reason for our much higher overall performance on the cell line data). Thus, there is a trade-off between controlling for cell type similarity and building imbalanced datasets with a larger number of cell types.

To address the reviewer's concern, we have nonetheless created additional imbalanced datasets composed of 500 cells each. These new datasets are composed of 400 cells of the

major cell type, with 25 cells from four additional cells each. The paired balanced dataset is composed of 100 cells of the same five cell types each (we have added a table summarising the cell types used for this analysis to the methods section of the manuscript). To address the reviewer's earlier comment as well in the context of this analysis we have also calculated the balanced accuracy, F1-score, Matthew's correlation coefficient, kappa and sensitivity values for this second imbalance analysis (S. Fig 28, 29). In addition, we also calculated the number of times a selection method is the top performing method or among the top three performing methods (S. Fig. 30). Overall, here we find that active learning outperforms other selection procedures.

S. Figure 28. Effect of expanded dataset imbalance on classification accuracy for the CyTOF - Bone marrow, scRNASeq - Breast cancer cell line and snRNASeq - Pancreas cancer datasets. Shown is the change for each metric (calculated as accuracy in imbalanced dataset - accuracy in balanced dataset / accuracy in balanced dataset). Each figure is faceted by cohort.

S. Figure 29. Effect of expanded dataset imbalance on classification accuracy for the scRNALung and tabula datasets. Shown is the change for each metric (calculated as accuracy in imbalanced dataset - accuracy in balanced dataset / accuracy in balanced dataset).

S. Figure 30. Number of times a selection procedure is the top performing method.

The average improvement score (as calculated for S. Figures 8 and 9) is calculated for each method, selection procedure, modality and metric. The number of times each selection procedure is the best performing (top) or among the best 3 performing (bottom) is shown for each metric. The selection procedures are ordered by the average number of times each method is among the top performing group.

Reviewer 2

Major

1. Are the proposed methods suited for transfer learning (building a classifier based on a prior dataset), or in the proposed methods, is there still the necessity of providing manual labels to a subset of cells from the different those found either through the active-learning process or unsupervised analyses? From my reading of the benchmark description at the bottom of page 4 (“These subsets were then used to train four cell type assignment methods using ground truth cell type labels”), it sounds like the previously designated ground truth labels were used in the training process, which may not reflect the typical process in which a user will need to actively define a new label for their newly collected dataset, given that no prior ground truth will be present for a new dataset. If the intended method can be easily applied to a scenario in which the user obtains a new, completely unlabeled dataset, the manuscript would benefit from demonstrating this (although perhaps only for a single use show-case of active learning, rather than sweeping through all related methods discussed).

We agree with the reviewer that transfer learning is frequently used to annotate single cell datasets. However, our approach is focussed on the scenario where no reference dataset exists. Specifically, we were interested in identifying the best way to select a set of cells to annotate manually that can then be used as a reference. Given the large number of experiments performed it was infeasible to manually annotate cells the way a user would, so we used existing labels to simulate this process. The ground truth cell type labels were used to simulate the user's manual annotations. We have added a note to our manuscript to clarify this point.

2. The specific methodology used for ‘known marker gene’ usage may not be appropriate, if I understand the approach correctly. On page 5, the authors state “To test this, we ranked all cells by the expression of a set of cell type marker genes provided by the dataset authors.” In essence, under this paradigm, we have ground truth labels, generated by the original authors, then gene-sets that were derived downstream of these cluster assignments through differential expression. The authors then use these known differentially expressed genes (as previously determined in this dataset), to identify the cluster labels that were used to perform the differential expression analysis that yielded the marker genes; this seems a bit logically circular. Can the authors clarify this point? If my original interpretation is correct, I believe that a better test of utilizing prior expected marker genes that would be more reflective of a real-world scenario would be utilizing previously published, orthogonally collected, marker genes from a marker gene database (so long as they are careful to ensure the datasets used for testing the method did not contribute to the marker gene database). The same comment may apply to other analyses if I have correctly understood the method as currently employed.

For the cell line data used we agree with the reviewer that our reasoning is circular. While “cell type” labels (i.e. cell line of origin) were known *a priori*, the marker genes for these cell lines were determined through a differential expression analysis between cell lines. This was a necessary step as no established marker genes for the cell lines used exist to our knowledge. Moreover, we would like to note the existence of a tradeoff for this part of our analysis: while well validated marker genes exist for “real” datasets, no ground truth cell type

labels exist. For cell line data on the other hand, ground truth “cell types” exist, but no cell type markers.

For all other data used in our study however, there is less circularity. For example, in the CyTOF dataset, a set of well established markers was used to assign cell types using a gating strategy, rather than doing a differential expression to identify these markers based on clusters. Similarly, in the snRNASeq dataset, tumour cells were identified using copy number profiles, thus not relying on markers at all, while all other cells were identified by clustering the cells and then identifying which markers were expressed in what clusters.

To further address this comment, we identified an independent set of markers for the liver and vasculature datasets by finding markers from the PanglaoDB database (Franzén et al., 2019) for each cell type in the tabula datasets. We have clarified this point in both the methods and main text of our manuscript.

Finally, we would like to note that the set of markers used only has an effect on two aspects for the active learning approaches: 1) the features used in the newly implemented SVM-rejection method (as this is how it was originally implemented by the authors (Abdelaal et al., 2019)) and 2) the initial set of cells selected. While we show that ranking the initial set of cells based on their marker expression is better than random selection, issues with the marker file will at worst make the cell selection process take longer.

To elucidate the effect of marker corruption on adaptive reweighting, we generated new marker files with 10, 25, 50, 75 and 100% of genes corrupted to a randomly selected gene not in the original marker file that was also among the 10,000 most highly expressed genes. Next, we applied the marker aware adaptive reweighting method and measured the impact of marker corruption in this cell type selection procedure method on the accuracy of the downstream cell type classification methods. Overall, there was a decrease in accuracy for most datasets, highlighting the importance of accurate marker files (S. Fig. 10).

S. Figure 10. Effect of adaptive reweighting marker corruption on sensitivity. Shown is the median sensitivity for each method and dataset as the markers used to select the initial cell population are increasingly corrupted from 0 to 100%.

3. On page 7, the authors state that “adaptive reweighting outperforms average selection and active learning across all accuracy measures and datasets, though with high variability (Fig. 3B).” It is a bit difficult to tell from the provided plots, given this variability. Could the authors please provide something like an average rank performance across all the tested datasets for each measure to more clearly display this?

We agree with the author that this was hard to interpret, especially now that we have doubled the number of datasets. We have visualised this more clearly in our updated manuscript (Fig. 3B). We note now with the expanded set of datasets that we no longer find a single method performs best in all settings, with either a variant of active learning or adaptive reweighting typically performing best for a given dataset.

Figure 3: Investigating the effect of different cell selection methods on predictive performance. B Performance of all selection methods tested across ten different train test splits (AL: active learning; AR: adaptive reweighting). Each selection method is ranked by the median balanced accuracy and sensitivity across seeds and cell type assignment methods. For the active learning results, the initial cell selection was ranked and a random forest was used.

4. Similar to the above, it is a bit difficult to clearly see in Figure 4, that having 0 cells in training increases entropy. Perhaps the authors could provide a summary panel, with the effect of ground truth cell type regressed out, as this also seems to have a large effect. As an aside – I agree with the authors’ intuition that the cell-type specific effect of why a given type’s absence may have variable effects on classifier performance (as with the schwann/endothelial example).

We agree with the reviewer that this could be visualised in better ways. To address the concern, we calculated and visualised the median entropy for the cell types that were removed as well as all cell types retained in the training dataset separately. We repeated this for each dataset and show that there is generally a large decrease in entropy upon addition of even just one cell, especially when using the logistic regression classifier. Overall, this summary panel is well in line with all our previous claims (Fig. 4A).

Figure 4: Unlabelled cells have higher entropy values. A Median scaled entropy values for each cohort when all cells of a type were removed (purple) and all other cells (yellow). Entropy of the cell type predictive distribution for all cells not in the initial training set of 20 cells. Boxplots are coloured by the number of cells present of a particular type (shown in the plot title), while the x axis shows the ground truth cell type label.

Minor

1. There may be an incomplete sentence, or extra word on page 5: "..., and would therefore benefit most from receiving a label of." Perhaps the word "of" should be removed?
2. In the legend of Fig. 2, the word size appears twice in close succession: "The cell number specifies the total size of the training set size."

We thank the reviewer for the thorough read of our work, both of these spelling issues have been fixed.

Reviewer 3

1. The current pipeline is only tested on 3 datasets and 5 cell type classification methods. In contrast, the conclusion and contribution made by this paper is the claim that some techniques are useful in general, rather than proposing a new method. Therefore, I think it is necessary to benchmark on more cell type assignment methods and datasets.

We apologise for the confusion in regard to this point. We would like to note that we do propose a new method (adaptive reweighting), though this is used to select cells for labelling rather than predict cell types. In addition, we would like to emphasize that our work is not predominantly concerned with identifying the best cell type assignment algorithm. Rather, we are measuring the improvement that can be obtained for *any* cell type assignment method by selecting a training dataset more intelligently.

Overall, we have had to make tradeoffs between benchmarking breadth (number of parameters explored and cell type *selection* methods used, see the table below we have added to our manuscript) and benchmarking depth (number of datasets and cell type *assignment* methods). In addition, this benchmarking is constrained by the computational time required (~4 weeks on a highly parallelized HPC) and the availability of ground truth data to minimise circularity in evaluation. Nonetheless, to address this limitation, we have doubled the number of datasets used in our benchmarking and added two additional cell type assignment methods (see below).

2. More cell type assignment methods:

To expand the number of cell type assignment methods, we have implemented two additional methods from two different model classes: a support vector machine and SingleCellNet as these were found to be the top performers in a benchmarking study using 27 datasets and 22 cell type assignment methods (Abdelaal et al., 2019). We selected SingleCellNet rather than scPred (the second-best performing method) since the latter was the top performing method not based on a support vector machine, while the former was another support vector machine based method.

3. More datasets. I think it is necessary to test on datasets with more cell types, including Tabula Muris, Tabula Drosophilae, Tabula Muris Senis, and Tabula Sapiens.

We agree with the reviewer that more datasets would help to further improve our study. To this end we have taken the reviewers suggestion and doubled the number of datasets. We incorporated the vasculature datasets from the Tabula Sapiens project and single cell data from the liver atlas. Since ground truth labels for these datasets are not available, we additionally added another dataset from five lung cancer cell lines and used the cell type of origin as a cell type label. Overall, our study is now composed of twice the number of datasets.

	Parameter	Possible values	Number of configurations tested
Models			
Scmap	Classification level	 ● Cell level ● Cluster level 	2
SingleR	Default		1
RandomForest	Internal grid search over parameters		1
CytoTOF-LDA	Default		1
SVM-rejection	As described in (Abdelaal et al., 2019)		1
SingleCellNet	Default		1
Subtotal			7
Cell selection			
Active learning	Initial cell selection	 ● Random ● Ranking 	2
	Entropy uncertainty	 ● Highest entropy ● 95% entropy ● 75% entropy 	3
	Maxp uncertainty	 ● Lowest maxp ● 5% maxp ● 25% maxp 	3
Subtotal			12*
Adaptive reweighting	KNN parameter	 ● 10 ● 20 ● 30 	3
	Clustering resolution	 ● 0.4 ● 0.8 ● 1.2 	3
Subtotal			9
Random selection	Repeated 3 times		3
Subtotal			3
Number of cells selected		100, 250, 500	3
Total			504**

* The total is 12, as for each entropy and maxp selection metric each cell selection is run, thus the total is $3 \times 2 + 3 \times 2 = 12$.

** The total number of jobs per dataset split is the number of cell selection methods times the number of models fit times the number of cells selected: $((12 \times 7) + (9 \times 7) + (3 \times 7)) \times 3 = 504$.

Supplemental table 1. Shown is the total parameter space explored for a single train-test split and modality of our benchmarking pipeline.

4. Evaluate the cross dataset setting. Real-world cell type assignment is often in a cross-dataset setting, where the model is trained on one dataset and then used to classify cells in another independent dataset. This would involve batch correct and mapping of cell types. The noise in marker genes and annotated labels in the training data might also affect the performance on the test dataset. It is therefore necessary to evaluate this setting.

We agree with the reviewer that transfer learning is frequently used to annotate single cell datasets. However, our approach is focussed on the scenario where no reference dataset exists. Specifically, we were interested in identifying the best way to select a set of cells to annotate manually that can then be used as a reference. Given the large number of experiments performed it was infeasible to manually annotate cells the way a user would, so we used existing labels to simulate this process. The ground truth cell type labels were used to simulate the user's manual annotations. We have added a note to our manuscript to clarify this point.

5. Could you comment on the running time of using these additional techniques? Will that make the training time substantially longer?

In designing this study, we took runtime into account from the beginning, as it is critical for the user to get the next set of cells quickly. As such, we implemented random forest and logistic regression based active learning setups, both of which are light-weight machine learning methods.

S. Figure 12. Runtime analysis for all selection methods benchmarked. Shown is the runtime in seconds for each selection method and dataset. All active learning methods were trained using a random set of 20 initial cells.

To quantify runtime, we also benchmarked the time required for each method to select the initial set of cells (S. Fig 12). With a median range of between 2-25 seconds across all

methods and modalities no selection method is too slow for practical purposes. Furthermore, significant runtime improvements can be made to the active learning setups as our implementation was not designed for speed. In addition, production level implementations of these approaches should be designed in such a way that the active learning classifier runs while the user is annotating a set of cells. Even with runtimes as slow as in our implementation, this should be fast enough for the bottleneck to be user annotation speed rather than active learning runtime.

Finally, we would like to note that there is a difference between adaptive reweighting as proposed in this work and active learning: In the latter case, the classifier has to be re-trained with every new batch of cells. Adaptive reweighting on the other hand only runs once at the beginning, thus continuous runtime for adaptive reweighting is less of a concern.

6. Noise in the cell type marker file. Cell type marker genes could be noisy. It might be interesting to randomly add noise to the makers and see if the proposed method is robsut to such noise.

We agree with the reviewer that this is an interesting and important experiment. To address this, we generated new marker files with 10, 25, 50, 75 and 100% of genes corrupted to a randomly selected gene not in the original marker file that was also among the 10,000 most highly expressed genes. Next, we applied the marker aware adaptive reweighting method and measured the impact of marker corruption in this cell type selection procedure method on the accuracy of the downstream cell type classification methods. Overall, there was a decrease in accuracy for most datasets, highlighting the importance of accurate marker files (S. Fig. 10).

S. Figure 10. Effect of adaptive reweighting marker corruption on sensitivity. Shown is the median sensitivity for each method and dataset as the markers used to select the initial cell population are increasingly corrupted from 0 to 100%.

References

- Abdelaal, T., Michielsen, L., Cats, D., Hoogduin, D., Mei, H., Reinders, M. J. T., & Mahfouz, A. (2019). A comparison of automatic cell identification methods for single-cell RNA sequencing data. *Genome Biology*, *20*(1), 194.
- Franzén, O., Gan, L.-M., & Björkegren, J. L. M. (2019). PanglaoDB: a web server for exploration of mouse and human single-cell RNA sequencing data. *Database: The Journal of Biological Databases and Curation*, 2019.
<https://doi.org/10.1093/database/baz046>
- Geuenich, M. J., Hou, J., Lee, S., Ayub, S., Jackson, H. W., & Campbell, K. R. (2021). Automated assignment of cell identity from single-cell multiplexed imaging and proteomic data. *Cell Systems*, *12*(12), 1173–1186.e5.
- Ren, P., Xiao, Y., Chang, X., Huang, P.-Y., Li, Z., Gupta, B. B., Chen, X., & Wang, X. (2021). A Survey of Deep Active Learning. *ACM Comput. Surv.*, *54*(9), 1–40.

Reviewer #2 (Remarks to the Author):

The authors have largely addressed my concerns, particularly in creating the new graphics. I believe the manuscript would benefit from a small discussion (without additional experiments) on a few points, that could help bring a more full context to these approaches when applied in broader practice.

1) The benefits and limitations of labeling individual cells vs groups of cells (as is often done in standard transfer learning, where an entire cluster is given the same label). The pros and cons of this come from the same source, borrowing information from neighbor cells in the graph. The major benefit of labeling groups of cells is blunting dropout of marker genes, which labeling single cells will be susceptible to. The pitfall of giving groups of cells the same label, which the procedure described here does not have, is when heterogeneity exists within a cluster, these may be more properly annotated.

2) How labeling individual cells may provide an advantage, particularly in the context of batch corrected analyses, in which heterogeneous populations can be merged into a single apparent cluster, which if given the same label can lead to inaccuracies.

Additional notes for the author's consideration (not to be considered for this manuscript's acceptance):

It may be interesting to explore a hybrid approach related to the above comments, consisting of using more granular 'meta-cells' for the manual labeling phase, giving these 10 or so cells the same label for the classifier to blunt the effect of dropout. This may also prevent there from being too much heterogeneity among the manually labeled cells.

Reviewer #3 (Remarks to the Author):

I believe the authors have addressed all concerns raised by reviewer 1. As described in the author response, although this manuscript did not propose a new cell type classification method, adaptive reweighting can be used to boost the performance of existing cell type annotation pipelines. Therefore, I think this manuscript could substantially contribute to the community.

Reviewer #2 (Remarks to the Author):

The authors have largely addressed my concerns, particularly in creating the new graphics. I believe the manuscript would benefit from a small discussion (without additional experiments) on a few points, that could help bring a more full context to these approaches when applied in broader practice.

1) The benefits and limitations of labeling individual cells vs groups of cells (as is often done in standard transfer learning, where an entire cluster is given the same label). The pros and cons of this come from the same source, borrowing information from neighbor cells in the graph. The major benefit of labeling groups of cells is blunting dropout of marker genes, which labeling single cells will be susceptible to. The pitfall of giving groups of cells the same label, which the procedure described here does not have, is when heterogeneity exists within a cluster, these may be more properly annotated.

2) How labeling individual cells may provide an advantage, particularly in the context of batch corrected analyses, in which heterogeneous populations can be merged into a single apparent cluster, which if given the same label can lead to inaccuracies.

Additional notes for the author's consideration (not to be considered for this manuscript's acceptance):

It may be interesting to explore a hybrid approach related to the above comments, consisting of using more granular 'meta-cells' for the manual labeling phase, giving these 10 or so cells the same label for the classifier to blunt the effect of dropout. This may also prevent there from being too much heterogeneity among the manually labeled cells.

We thank the reviewer for these additional comments. We have added a section to our discussion (pages 9-10 of revised manuscript) to address the pros and cons of labelling individual cells compared to cell clusters with respect to cluster heterogeneity and dropout effects, and incorporate the reviewer's suggestion of labelling metacells.

Reviewer #3 (Remarks to the Author):

I believe the authors have addressed all concerns raised by reviewer 1. As described in the author response, although this manuscript did not propose a new cell type classification method, adaptive reweighting can be used to boost the performance of existing cell type annotation pipelines. Therefore, I think this manuscript could substantially contribute to the community.

We thank the reviewer for reviewing our revised manuscript as well as the comments by reviewer 1.